# Investigating the Association between Streetscapes and Mental Health in Zhanjiang, China: Using Baidu Street View Images and Deep Learning

**DOI:** 10.3390/ijerph192416634

**Published:** 2022-12-11

**Authors:** Anjing Zhang, Shiyan Zhai, Xiaoxiao Liu, Genxin Song, Yuke Feng

**Affiliations:** 1Key Laboratory of Geospatial Technology for the Middle and Lower Yellow River Regions, Henan University, Ministry of Education, Kaifeng 475004, China; 2Department of Geography, University of Zurich, 8057 Zurich, Switzerland; 3College of Geography and Environmental Science, Henan University, Kaifeng 475004, China; 4Department of Community Health Science, Cumming School of Medicine, University of Calgary, 2500 University Drive NW, Calgary, AB T2N 1N4, Canada

**Keywords:** street view image, streetscapes, deep learning, mental health

## Abstract

Mental health is one of the main factors that significantly affect one’s life. Previous studies suggest that streets are the main activity space for urban residents and have important impacts on human mental health. Existing studies, however, have not fully examined the relationships between streetscape characteristics and people’s mental health on a street level. This study thus aims to explore the spatial patterns of urban streetscape features and their associations with residents’ mental health by age and sex in Zhanjiang, China. Using Baidu Street View (BSV) images and deep learning, we extracted the Green View Index (GVI) and the street enclosure to represent two physical features of the streetscapes. Global Moran’s I and hotspot analysis methods were used to examine the spatial distributions of streetscape features. We find that both GVI and street enclosure tend to cluster, but show almost opposite spatial distributions. The Results of Pearson’s correlation analysis show that residents’ mental health does not correlate with GVI, but it has a significant positive correlation with the street enclosure, especially for men aged 31 to 70 and women over 70-year-old. These findings emphasize the important effects of streetscapes on human health and provide useful information for urban planning.

## 1. Introduction

Many countries have experienced dramatic urbanization, and more than half of the world’s population now resides in urban areas [1,2]. With the rapid advancement of the new urbanization policy, China’s urbanization rate has increased to 60.6% in 2019 [3]. At the same time, a series of problems have been brought about by urbanization, such as housing shortages [4], lack of resources [5], air and water pollution [6], Chronic diseases [7], mental health problems [8], and so on. Some of the literature has indicated that rural-to-urban migrants are likely to suffer from more mental health problems, due to the changes in living environment, housing conditions, and perceived neighborhood safety [6,9]. The proportion of the Chinese population with a burden of mental, neurological, and substance use disorders increased by 20% from 1990 to 2013 and is projected to increase by another 10% by 2025 [10]. Therefore, understanding the relationship between the urban environment and mental health is necessary for urban designers and managers to improve urban livability for human health and welfare.

Both street greenness and street enclosure have been recognized as two important features of the urban streetscape, influencing the perception of the urban environment and the mental health of humans [11,12]. Urban street greenness refers to various forms of vegetation in the street, which has been considered one of the important landscape design elements in cities [13]. It is usually measured by the Green View index (GVI) in previous studies [14,15,16,17]. Many papers have pointed out that city street greenness contributed to the engagement of outdoor activities probability for residents, especially walking, which would benefit their mental health [16,18]. In addition, it was found that urban greenness could help people reduce stress and improve restorative quality [19,20,21], which was beneficial for residents’ mental health. Additionally, street greenness provides places for people to interact with each other and enhance social cohesion, which is good for mental health [22,23]. Ta et al. [24] found that small green space along the road contributes to individuals’ travel satisfaction and well-being. Yue et al. [25] showed that streetscape grasses have a stronger association with the mental health of older adults than street trees.

The street enclosure refers to the room-like spatial sense that is visually created on the street by buildings, trees, and other vertical elements [26]. It indicates how much sky is obstructed by buildings, and tree canopies [27]. On the one hand, the urban street enclosure space may affect the thermal comfort of pedestrians as the urban thermal environment is influenced by the geometry of the street including vegetation cover [28]. Previous papers have shown that the thermal comfort of cities influences the choice of outdoor activities and the use of urban space by urban residents [29,30]. Li et al. [31] found that human walking activity was influenced by street enclosure, with a significant positive correlation with the enclosed space formed by buildings and a negative correlation with that provided by trees on the recreational land. On the other hand, the street enclosure can be psychologically oppressive for pedestrians to some extent. Asgarzadeh, Koga [11] found that high-rise buildings are more oppressive than low-rise buildings, while street trees significantly reduce the oppressiveness of pedestrians in recreational land. Baran et al. [32] indicated that women are easier to perceive unsafe than men in high-enclosed and medium-enclosed environments, and the perceived safety can affect both physical and mental health outcomes. We thus take these two streetscape features to study whether there are relationships between the streetscape features and people’s mental health.

Previous studies mainly used questionnaire-based or GIS-based approaches to measure streetscape features at the macroscopic scale, which did not fully and objectively reflect the urban environment at the street scale and from the eye-level perspective [33]. However, with advances in computer vision, it is possible to measure street-scale features objectively by street view images. Google Street View (GSV) images, introduced in 2007, are images taken by a car along a street to simulate a pedestrian’s perspective. Many studies have indicated the feasibility of using street View Images to study street-scale environmental features [15]. Rundle et al. [34] compared seven environmental variables extracted through field surveys and Google GSV for 37 New York City neighborhoods. The results showed a high degree of consistency in 54.3% of the items. Zhang et al. [35] used street view images and deep learning to investigate the relationship between street visual elements and people’s perceptions. However, GSV cannot be obtained in China because of strategic and business policies. We thus used Baidu Street View (BSV), a GSV-like product developed by the Chinese company Baidu [36].

In summary, we find that there are still two shortcomings in the current research. First, the traditional methods using questionnaire- or GIS-based data to measure the features of the macro-scale streetscape, cannot fully examined the eye-level streetscape features. Second, most previous studies have focused on the relationship between green space and mental health, but research on the relationship between street enclosure and mental health is rare. The relationships between streetscape features and people’s mental health are still poorly understood. To address these research gaps, this study, takes Zhanjiang, Guangdong Province, as a case study. The aims of this study are as follows: (1) extracting two streetscape features, namely GVI and the street enclosure, using BSV images and a deep learning method, and (2) exploring the relationship between the features of urban streetscape and people’s mental health by age and sex. Our results may be helpful to understand the situation of the existing streetscape and provide suggestions for future urban design.

## 2. Materials and Methods

### 2.1. Study Areas

The study area is located in Zhanjiang City, between 20°12′ and 21°35′ N latitude and 109°31′ and 110°55′ E longitude, with a total population of around 7.0 million in 2020. With its tropical monsoon climate, the average annual temperature is about 23° and the coldest monthly average temperature is 17.2°. Due to the high aesthetic quality of the urban green spaces, Zhanjiang was declared a “National Garden City” by the Chinese Ministry of Construction in 2005 [37]. In 2020, the gross domestic product (GDP) was about 6.46 trillion yuan (RMB) [38]. With rapid economic development and population growth, Zhanjiang has experienced rapid urban expansion and large areas of agricultural land have been converted to nonagricultural and urban land [39]. At the same time, the Zhanjiang government has added mental diseases into their health insurance coverage to support the development of local mental health services since 2015 [40]. The study mainly covers four municipal districts of Zhanjiang, namely Chikan District, Xiashan District, Potou District, and Mazhang District (Figure 1).

### 2.2. Data

**Street View Images Data** To measure the urban streetscape features, we use street view images from Baidu Map in 2020, one of the most popular web-based maps in China [41]. It provides an open map platform service, through which users can acquire the BSV images at different street sites [42]. First of all, the road network of Zhanjiang City was downloaded from the Open StreetMap website. Before laying the sample points, this study used ArcGIS 10.6 to extract the centerline of all two-way roads into one-way roads. In terms of the spacing of sample points, to make each sample point’s field of view as interconnected and non-overlapping as possible, combined with the existing literature [43], this study chose to set a sample point every 200 m along the road network, resulting in a total of 11,916 sample points. We obtained street view images of each sample as panorama images from the public API interface of Baidu Maps (https://map.baidu.com/ accessed on 20 December 2020). Based on these points, only 3780 panorama images were obtained. From Figure 2, we found that most samples are located in Chikan District and Xiashan District which is the central part of Zhanjiang. It may be because the information on the Baidu map is only available in the city center. We built a proxy for exposure with four distances (200 m, 500 m, 750 m, and 1000 m) to streetscape according to previous studies about environmental characteristics [44,45] and the sample distance in this study.

**Mental health data** Resident’s mental health was assessed using hospitalization data of patients suffering from psychiatric disorders in general hospitals from 2014 to 2015 in Zhanjiang City, Guangdong Province, with a total of 813 patients. The disease data were provided by Guangdong Medical University and included each patient’s admission time, place of residence, and primary diagnosis. According to the International Classification of Diseases, 10th Revision [46], the primary diagnosis mainly included psychiatric disorders such as organic mental disorder, recurrent depressive disorder, partially undifferentiated schizophrenia, obsessive-compulsive schizophrenia, schizoaffective disorder, acute schizophrenia-like psychotic disorder, epileptic psychosis, mental retardation, and psychotic disorder. We found that 15 patients under were 5 years old. We removed these cases as we thought the effects of urban streetscape could be ignored. Finally, there were only 798 patients left, with 410 males and 388 females. There are 323 patients aged 11 to 30, 302 patients aged 31 to 50, 135 patients ranging from 51 years old to 70, and 38 patients aged between 70 and 90. Figure 3 shows the number of patients by age and sex. It was found that 119 patients live in Chikan District, 267 patients live in Xiashan District, 151 patients live in Potou District, and 280 patients live in Mazhang District.

### 2.3. Methods

**Framework design** Figure 4 shows the framework of this study. First, we obtained a panoramic street view image of each sample point from Baidu Maps, every 200 m along the road. Second, to extract SVF, panorama images were required to convert to fisheye images first. We then carried out the semantic segmentation of panorama images to extract GVI and the semantic segmentation of fisheye images to extract street enclosure using a deep learning method. We also obtained the addresses of 817 patients with mental health problems from Guangdong Medical University. Third, we divided the study area into small areas using the buffer method. We then measured GVI, street enclosure, and the level of mental health by age and sex for an area. Finally, we analyzed the spatial distribution characteristics of GVI and street enclosure using Global Moran’s I and hotspots analysis. We also assessed the association between mental health and street features through Pearson correlation analysis.

**Green View Index (GVI)**: Yang et al. proposed an improved “Green View” index to evaluate the visibility of urban greenness. They defined GVI as the ratio of the number of pixels with vegetation per image [17]. This method has been used in many studies to calculate street greenness in many studies [15,17].

**The street enclosure:** The street enclosure is defined as the ratio of non-sky pixels in an image. According to Li et al., it can be calculated by using 1 minus Sky View Factor (SVF) [31,47]. As a dimensionless parameter of urban geometry, the Sky View Factor (SVF) indicates the degree of obstruction of the sky by buildings and trees. The SVF is zero when the sky is completely obstructed and 1 when the SVF is free of obstructions [31]. To extract SVF, panorama images were required to convert to fisheye images first. In this study, the panoramic image was converted into a fisheye image by PTgui (a software for processing images), and the process is shown in Figure 5. In this paper, we would use the enclosure of the street caused by both buildings and trees.

**Image Semantic Segmentation Based on Deep Learning** This study used a Fully Convolutional Network for semantic image segmentation (FCN), which can predict each pixel’s semantic property in an image [48]. It has been widely used in some studies of street view image classification, including the recognition and extraction of elements such as sky, vegetation, and buildings for streetscape images [14,47]. We used a model built by Yao et al. [49]. A detailed description of this model can be found in Yao et al.’s study [49]. We used this model to extract the proportions of greenness and sky in the street view images.

**Global Moran’s I and Hotspot Analysis** To examine the spatial distribution of the GVI and the street enclosure, Global Moran’s I and hot spot analysis in ArcGIS were used to test the spatial dependence of the objects. For Moran’s I index, the significance is calculated by *p* values and Z scores. If significant, a positive Moran’s I index value indicates a positive relationship between the variables and their cluster distribution. A negative Moran’s I index value indicates a negative relationship between the variables and the scattered distribution of these variables. If not significant, the distribution is random and there is no relationship between the variables [50]. The hot spot analysis tool in ArcMap identifies a significant hot/cold spot based on the attribute values of its neighbors. A significant hotspot is a feature with a high attribute value surrounded by other adjacent features that also have high values [51]. When implementing these methods in ArcGIS, an appropriate spatial relationship between features is needed to reflect the spatial and distributional circumstances of actual target features. Several different conceptualizations for spatial relationships are available, such as inverse distance, fixed distance band, and zone of indifference. For both methods, we used a fixed distance band which is a distance preset by the tool that determines which neighbors are to be included in the analysis. Previous research shows that it is appropriate to analyze point datasets [52].

**Correlation analysis** Firstly, grids of 200 m, 500 m, 750 m, and 1000 m size were created in ArcGIS in the study area to simulate the buffer zone at different distances. The grids data were merged with the disease data and the street sample point data, and only the grids with both street sample points and disease cases were retained. With the address of each patient, the number of patients within each grid was aggregated at different grid sizes by sex and age. It was used to represent the mental health status of residents within this certain area. It is notable that even if a patient could be hospitalized more than once, we only used the patient’s address once. The average values of GVI and street enclosure were calculated to represent the physical features of the street within each grid. Table 1 is the statistical analysis of two street features and the number of patients at different buffer distances. Finally, Pearson’s correlation was used to explore the relationship between residents’ mental health and these two streetscape features at different buffer distances.

## 3. Results

### 3.1. Spatial Patterns of GVI in the Whole Study Area

The average GVI for the sample sites in the study area is 7.03% (median: 4.79%, range: 0–53.54%). Figure 6 shows the histogram of GVI, suggesting that most sample sites have GVIs lower than 20%. Figure 7 reveals that there were a few high-value points of GVI, which are sporadically distributed in the study area, and most of the areas have a low GVI value. Global Moran’s I is 0.36 (z-score: 77.02, *p*-value: 0.00). The results demonstrate GVI values tend to cluster. As shown in Figure 8, the cold spot is widely distributed while the hot spot areas of GVI are patchily distributed. To be specific, there is one hot spot in the southwest Mazhang District, an obvious one in the central Chikan District, and a big one in the southern Xiashan District.

### 3.2. Spatial Patterns of Streets Enclosure in the Whole Study Area

The average value of street enclosure for sample sites is 54.77% (median: 54.72%, range: 22–94%). Figure 9 shows most values of enclosure of the street range from 40% to 70%. Figure 10 shows that the values of the street enclosure are relatively high in the urban center area, and low values are distributed in the periphery of the city. Global Moran’s I is 0.48 (z-score: 94.84, *p*-value: 0.00). The results indicate a statistically significant cluster of the street enclosure. Figure 11 indicates that the street enclosure had a different spatial pattern from the GVIs. The hot spot areas are mainly distributed in two major areas. One covers the whole area of Chikan District and northeast of Mazhang District, and the other locates in the east of Xiashan District and southwest of Potou District.

### 3.3. Analysis of the Association between Streetscape Features and Mental Health at Different Buffer Distances

Table 2 demonstrates that there is no significant correlation between the GVI and mental health in Zhanjiang. The street enclosure is not correlated with the mental health of residents at the 200 m but shows a significant positive correlation with the number of cases with mental health problems from the 500 m buffer zone onward. Remarkably, the correlation coefficient increases with the distance of the buffer zone.

Table 3 shows that starting from a buffer zone of 750 m, the mental health of men aged 31 to 70 years was positively associated with the street enclosure. This relationship only appears in women over the age of 70.

## 4. Discussion

This study explores the relationship between residents’ mental health and the physical features of streetscapes in the city of Zhanjiang. A large number of streetscape images are used to assess the street greenness and street enclosure. The results reveal that most values of GVI are lower than 20%, while all the values of the street enclosure are higher than 20%. Most samples with relatively low GVI are in the city center, while those with relatively high GVI are located in the urban periphery. The distribution of the street enclosure values is almost opposite to that of GVI. Previous studies suggest that GVI has a strong correlation with the canopy coverage close to sample sites, and the street enclosure is affected by the number and size of trees along streets and the heights of buildings [18,33]. The reason why the values of GVI are high in periphery areas and some parks is that trees tend to grow more abundantly there. For the distribution of the street enclosure, the large number of tall buildings and skyscrapers in the city center maybe be the main factor contributing to the high values of the street enclosure.

Pearson’s correlation analysis shows that there is no significant association between street greenness and people’s mental health. However, street enclosure has a significantly positive correlation with the prevalence of mental health problems. After grouping patients by sex and age, the results show that street enclosure is positively associated with the mental health of men aged 31 to 70, and women aged over 70. Our findings on the effects of street greenness are inconsistent with some studies [53,54], but consistent with these studies [55,56,57]. One possible reason might be that the degree of urbanization and income is not taken into account. Mitchell and Popham [58] suggest that in higher income suburban and rural areas, green space has no impact on residents’ health. Both the type and quality of green space rather than the quantity of green space [28,54,55] play an important role in the perception of the landscape. However, in this study, we do not consider these factors. In terms of the street enclosure, our study indicates that street enclosure contributes to the prevalence of mental health problems. Further, the mental health of 31–70-year-old men is affected by street enclosure, and this effect only occurs for women over 70 years old. Our findings are similar to those of Wang et al. [59], who measured walkability according to the visual enclosure of a neighborhood for assessing the mental health impacts. Their results showed that the proportion of sky contributed to the alleviation of anxiety and depression, especially for disadvantaged older adults.

This study provides an overview of Zhanjiang streetscapes using streetscape images and deep learning methods. As the complex impacts of urbanization on health have been increasingly recognized [60], our findings may provide useful information for the municipal government to design a more livable and healthier urban space. To be specific, future urban environmental management projects should consider how to lower the enclosure of city streets. For example, constructing low buildings and planting grass or shrubs instead of tall trees should be taken into account, which is helpful to encourage people to walk more and alleviate people’s stress and anxiety.

There are some limitations in this study. Firstly, we do not consider the balance of the available samples of the patients in the four districts. Second, the mental health of residents in an area is represented by the number of cases in an area by establishing grids. It may be better to use the incidence of psychiatric disorders to represent the mental health situation of residents. Third, there are many other features of the street such as neatness and pedestrian flow that may also affect the mental health status of residents. Future research should distinguish between street enclosure caused by trees and that caused by buildings, which may lead to different results. Furthermore, we did not take into account other social group characteristics such as education level, income, and the frequency and duration of street activities. To summarize, future studies should highlight these points (1) more streetscape characteristics and more social group characteristics should be considered; and (2) try to find other indicators to represent human mental health.

## 5. Conclusions

In this study, we measured two street features, GVI, and street enclosure, in Zhanjiang using streetscape images and a deep learning method. Combined with mental health data, we assessed the associations between two features of the streetscape and residents’ mental health. Our results indicate that GVI and street enclosure show an almost opposite spatial distribution trend in Zhanjiang. The street enclosure is correlated with residents’ mental health, in particular for men aged 31 to 70 years while GVI has no significant correlation with it. This study shows the spatial distribution of street greenness and street enclosure in Zhanjiang city, which provides reference materials for future city design and construction programs. It also suggests the municipal government keep the street enclosure as low as possible to protect the mental health of city dwellers when designing urban streetscapes.

## Figures and Tables

**Figure 1 ijerph-19-16634-f001:**
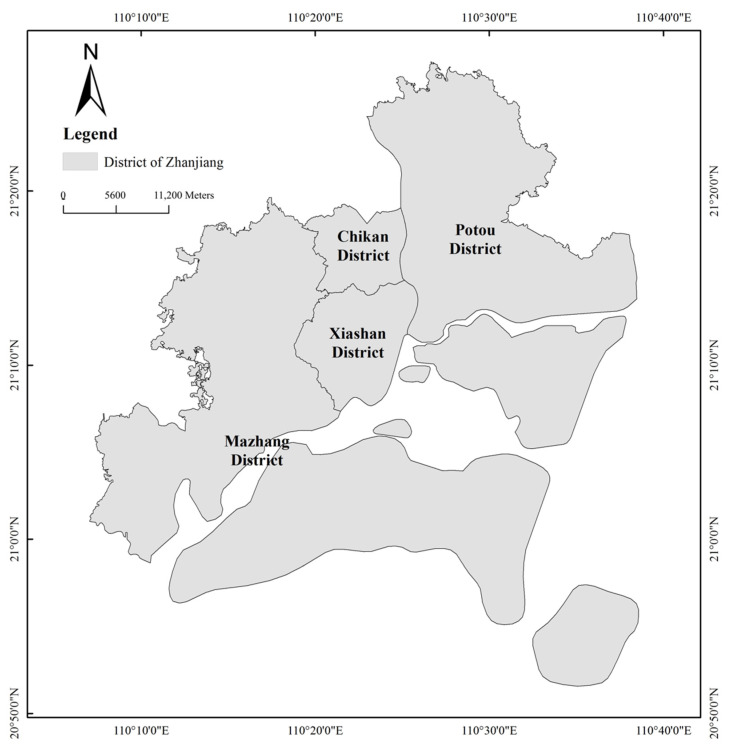
The map of four districts of the study area.

**Figure 2 ijerph-19-16634-f002:**
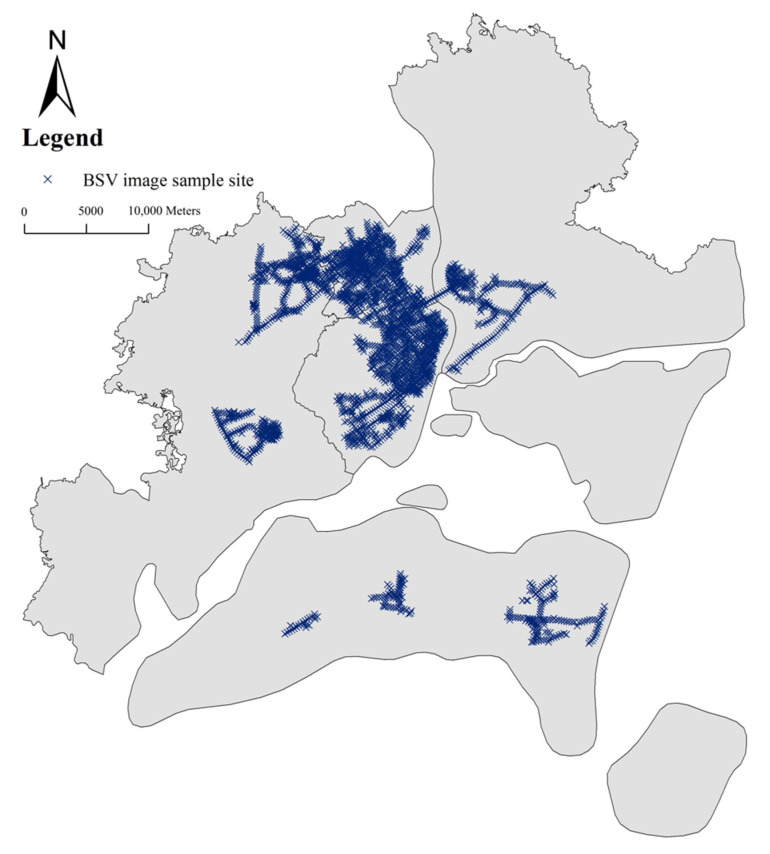
The sample sites of BSV image.

**Figure 3 ijerph-19-16634-f003:**
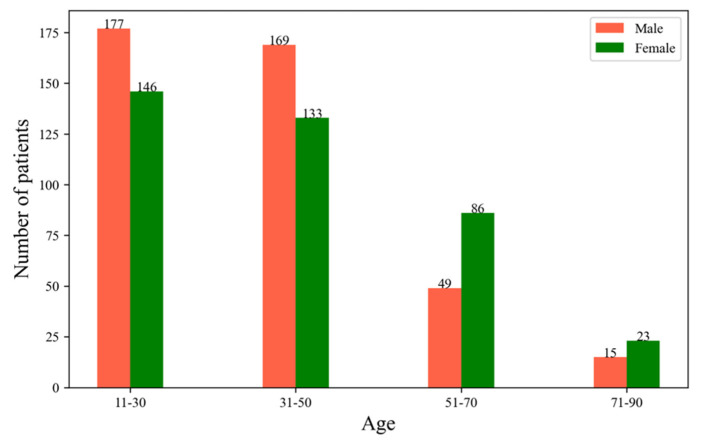
Number of patients in different age and gender groups.

**Figure 4 ijerph-19-16634-f004:**
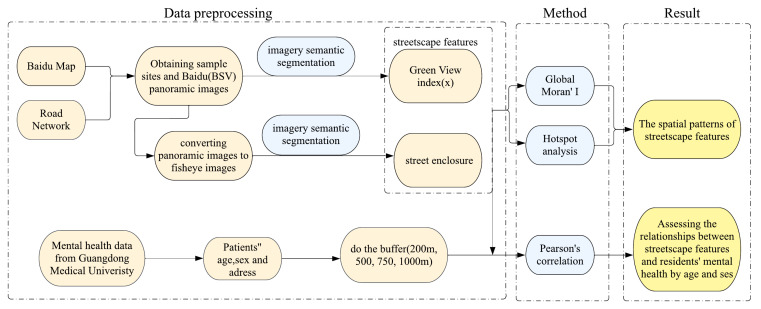
The framework for analyzing the spatial distribution of two streetscape features and their association with mental health. Note: the blue represents input data; the green represents methods; the orange represents the outputs.

**Figure 5 ijerph-19-16634-f005:**
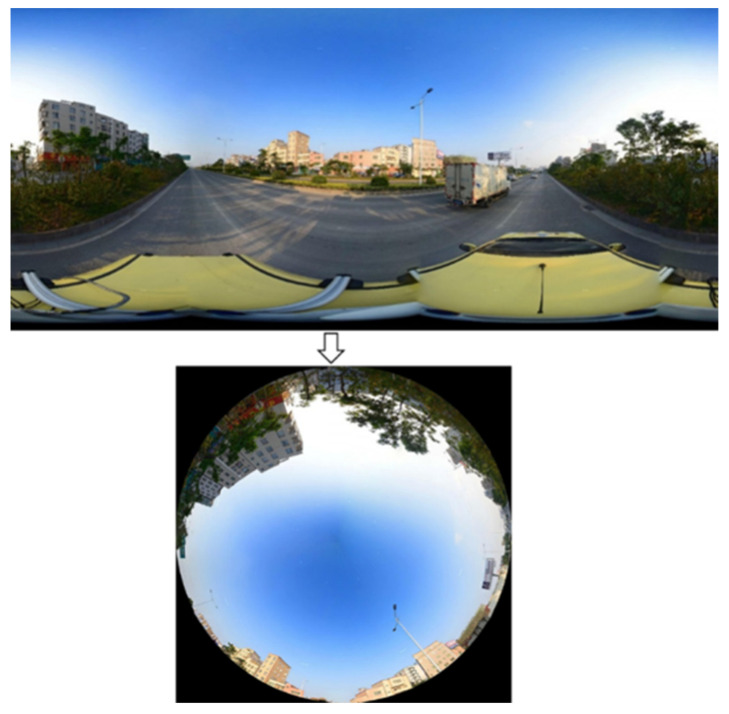
Converting panorama image to fisheye view image.

**Figure 6 ijerph-19-16634-f006:**
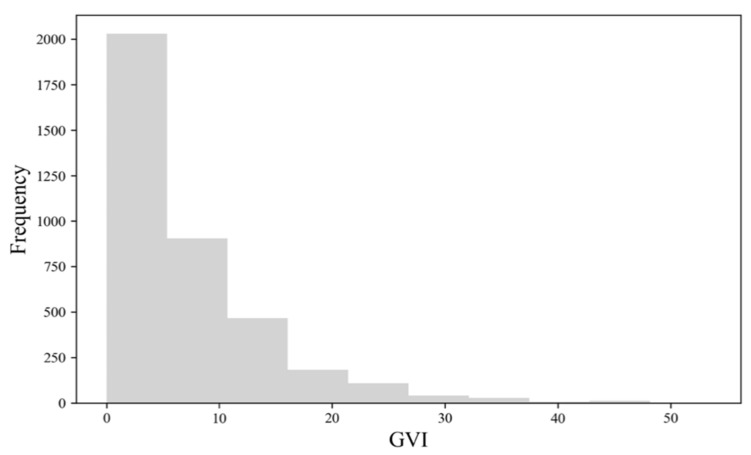
GVI histogram.

**Figure 7 ijerph-19-16634-f007:**
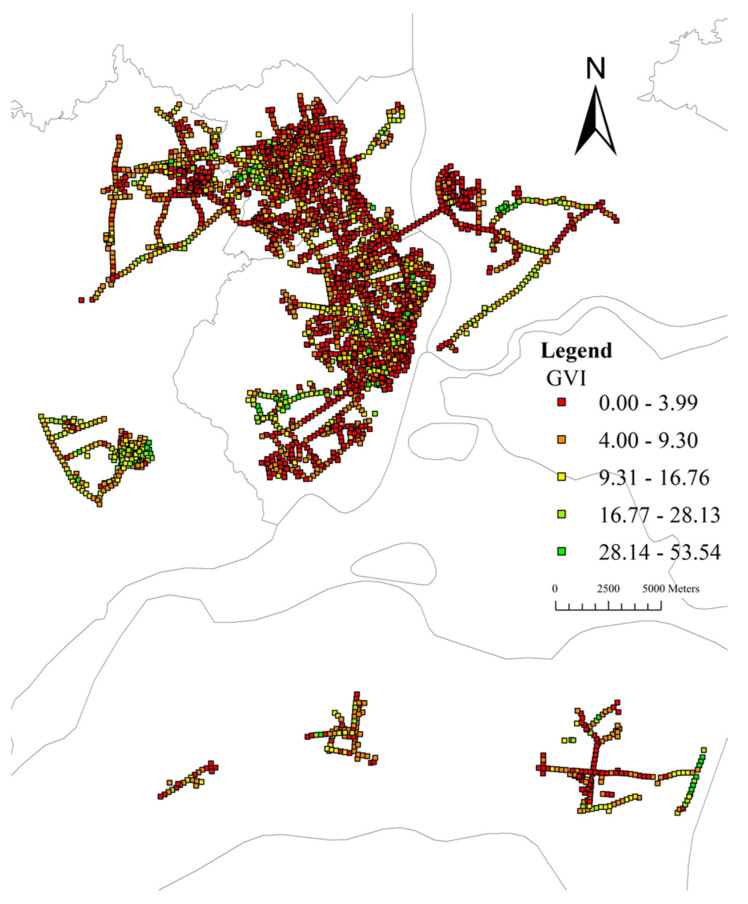
GVI spatial distribution map.

**Figure 8 ijerph-19-16634-f008:**
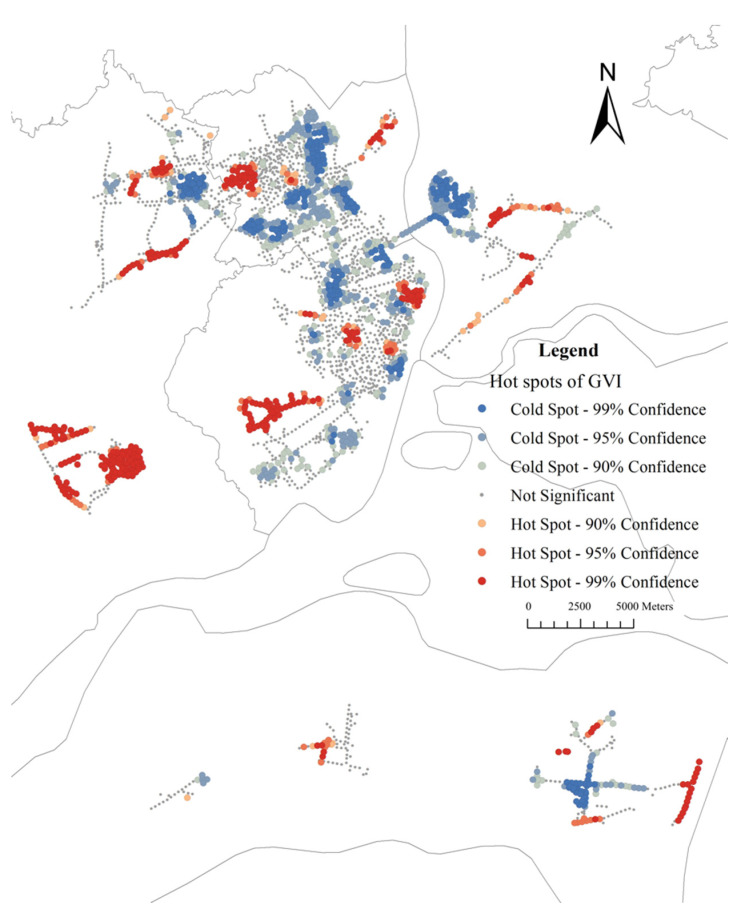
Distribution map of GVI hotspots.

**Figure 9 ijerph-19-16634-f009:**
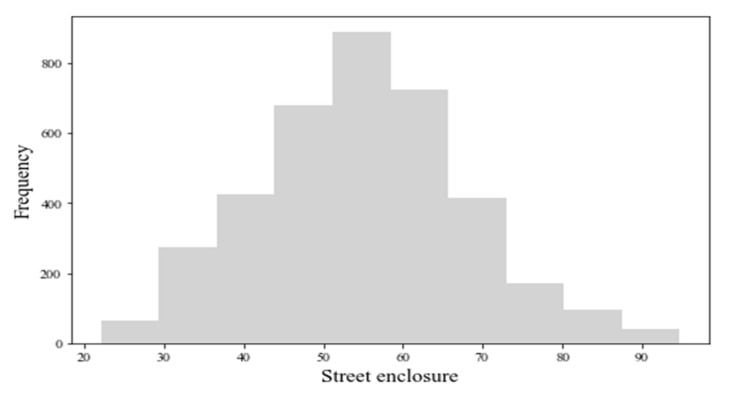
Street enclosure histogram.

**Figure 10 ijerph-19-16634-f010:**
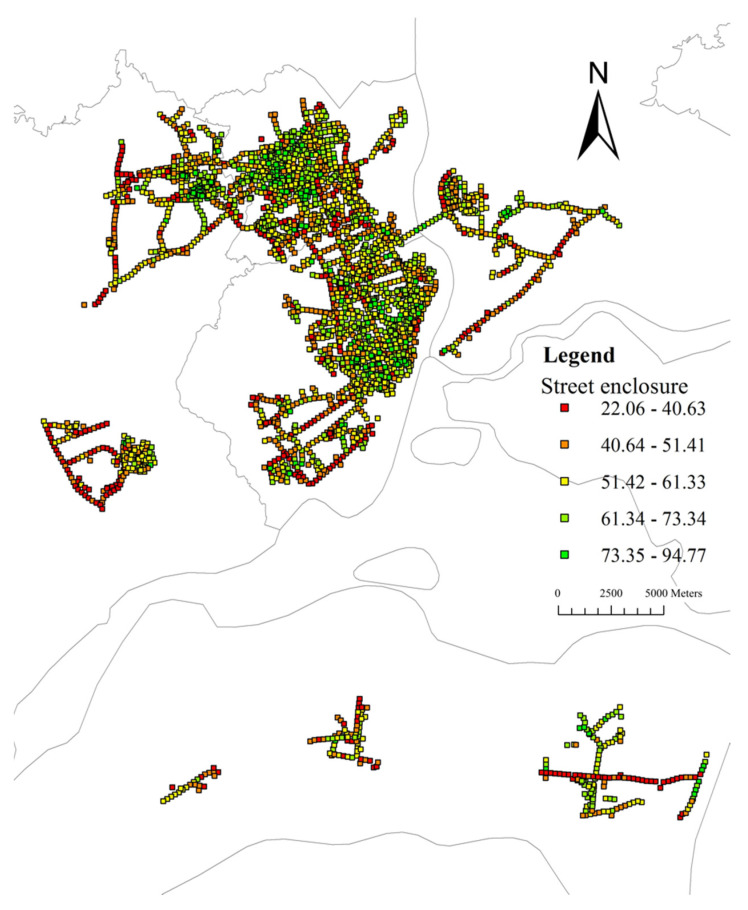
Street enclosure spatial distribution map.

**Figure 11 ijerph-19-16634-f011:**
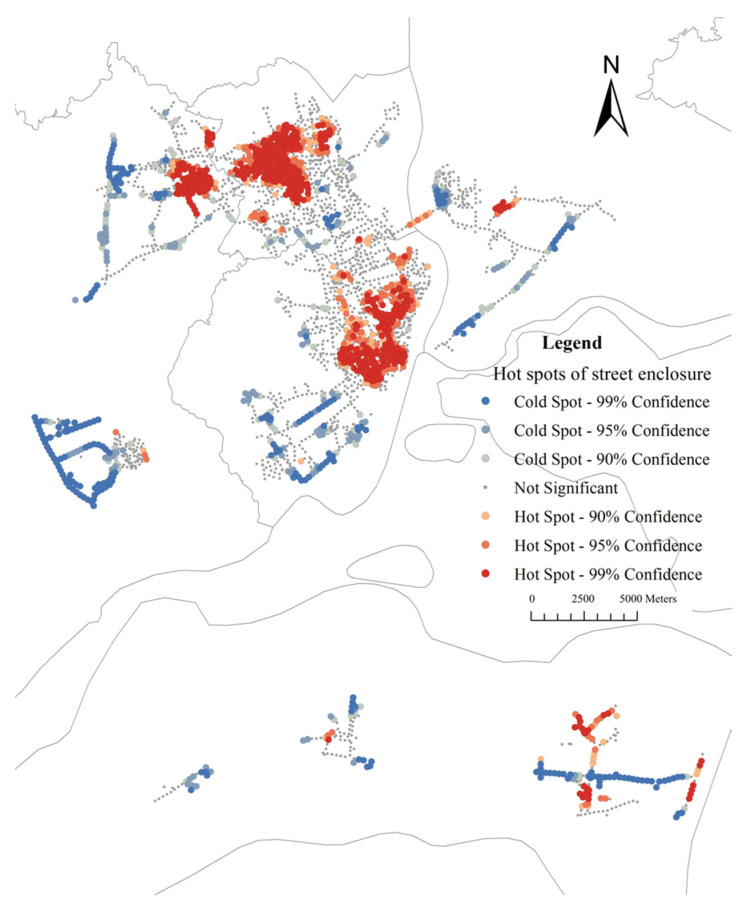
Distribution map of street enclosure hotspot.

**Table 1 ijerph-19-16634-t001:** Statistical analysis of streetscape features and number of whole patients at different buffer distances.

			Buffer Distance (Meter)			
			200	500	750	1000
whole patient	number of samples		211	160	127	112
	GVI	maximum	35.66	35.03	23.17	17.05
		minimum	0.00	0.60	0.09	0.39
		mean	6.28	6.83	6.24	6.33
		median	4.01	5.75	5.22	5.67
	street enclosure	maximum	91.89	78.76	73.98	70.34
		minimum	32.16	30.90	32.24	30.94
		mean	60.25	57.52	55.41	54.56
		median	59.24	58.14	56.68	54.77
	number of patients	maximum	8	13	28	39
		minimum	1	1	1	1
		mean	1.64	2.73	3.60	4.05
		median	1	2	2	2
patients aged 11–30	number of samples		82	86	71	71
	GVI	maximum	29.70	19.88	23.17	14.83
		minimum	0.00	0.60	0.09	0.39
		mean	6.22	6.51	6.39	6.21
		median	3.41	5.97	5.33	5.78
	street enclosure	maximum	90.78	78.76	73.98	70.34
		minimum	32.16	31.99	33.44	30.94
		mean	61.19	57.56	56.38	54.92
		median	60.84	57.20	57.20	56.72
	number of patients	maximum	5	5	8	12
		minimum	1	1	1	1
		mean	1.46	1.73	2.14	2.24
		median	1	1	2	2
patients aged 31–50	number of samples		94	88	79	63
	GVI	maximum	35.66	35.03	15.75	16.23
		minimum	0.00	0.60	0.73	0.39
		mean	6.30	6.48	5.90	6.49
		median	4.25	5.12	4.96	5.68
	street enclosure	maximum	91.89	78.76	73.98	70.34
		minimum	32.45	41.77	38.92	46.33
		mean	60.13	59.15	57.35	57.53
		median	58.45	58.14	57.77	57.30
	number of patients	maximum	4	7	12	18
		minimum	1	1	1	1
		mean	1.36	1.92	2.35	2.75
		median	1	1	1	2
patients aged 51–70	number of samples		59.00	62.00	49.00	45.00
	GVI	maximum	31.08	29.94	20.49	17.05
		minimum	0.00	0.98	1.09	1.66
		mean	5.67	7.25	6.50	6.51
		median	3.64	6.15	6.53	5.68
	street enclosure	maximum	90.01	78.76	69.03	68.73
		minimum	32.45	30.90	35.89	31.77
		mean	59.49	59.71	58.55	56.99
		median	59.75	61.00	58.78	59.41
	number of patients	maximum	8	11	12	16
		minimum	1	1	1	1
		mean	1.34	1.50	1.90	2.16
		median	1	1	1	1
patients aged 71–90	number of samples		16.00	17.00	17.00	14.00
	GVI	maximum	27.20	78.76	10.17	13.99
		minimum	0.00	0.98	2.81	3.85
		mean	8.03	21.30	6.47	7.19
		median	5.97	7.00	6.95	7.11
	street enclosure	maximum	86.52	68.37	69.03	64.83
		minimum	52.19	48.35	32.24	46.33
		mean	64.58	59.85	57.05	57.90
		median	63.30	58.52	60.27	58.78
	number of patients	maximum	4	2	4	5
		minimum	1	1	1	1
		mean	1.19	1.53	1.47	1.71
		median	1	2	1	1

**Table 2 ijerph-19-16634-t002:** Pearson correlation coefficients between GVI and enclosure and the number of cases at different buffer distances.

Buffer Distance (Meter)	GVI	Street Enclosure
200 m	−0.0582	−0.0088
500 m	−0.0413	0.2484 **
750 m	0.014	0.3201 **
1000 m	0.027	0.3906 **

** Correlation is significant at the 0.01 level (two-tailed).

**Table 3 ijerph-19-16634-t003:** Pearson correlation coefficients between GVI and enclosure and the number of cases at different buffer distances and in different sex and age groups.

Age	Sex	Street Features	Buffer Distance (m)			
			200	500	750	1000
11–30 years old	male	GVI	−0.1812	−0.1745	−0.1396	−0.0702
		street enclosure	−0.1047	0.0343	0.1042	0.1746
	female	GVI	0.0886	0.0694	0.1369	0.2141
		street enclosure	−0.3539	−0.01	0.2238	0.2821
31–50 years old	male	GVI	0.0762	0.0896	0.0846	−0.0153
		street enclosure	0.0404	0.0494	0.2815 *	0.3936 **
	female	GVI	−0.1911	0.1367	−0.0459	−0.0047
		street enclosure	−0.036	0.0132	−0.2354	0.1907
51–70 years old	male	GVI	−0.0147	−0.2487	−0.1823	0.0873
		street enclosure	−0.1004	0.322	0.4852 *	0.4286 *
	female	GVI	−0.0167	0.0091	0.0923	0.0582
		street enclosure	0.1042	0.1229	0.0461	0.2714
71–90 years old	male	GVI	NaN	0.6071	0.0614	0.1275
		street enclosure	NaN	0.273	0.3869	−0.1129
	female	GVI	−0.3087	−0.1526	−0.5332	0.0205
		street enclosure	0.0775	0.5477	0.5716	0.7057 *

** Correlation is significant at the 0.01 level (two-tailed); * Correlation is significant at the 0.05 level (two-tailed).

## Data Availability

The data in this study are available from the corresponding authors upon request. Due to the inclusion of sensitive personal information, these data are not publicly available.

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
