# Peer review of "Investigating the Association between Streetscapes and Mental Health in Zhanjiang, China: Using Baidu Street View Images and Deep Learning"

_ijerph, 2022, doi:10.3390/ijerph192416634_

Round 1

Reviewer 1 Report

This paper used Baidu Street view images and deep learning to analyze the association between streetscapes and mental health in Zhanjiang, China. The topic is interesting and fits the realm of the special issues. However, this paper has many issues to be cracked as following:

1. In the section of Abstract, the context of the association between streetscapes and mental health are suggested to be directly introduced. The limitation of existing studies should be explained. Also, results are overemphasized in the section, while the discussion and contribution of this paper are not introduced yet. Due to the main theme “the association between streetscapes and mental health” entitled in this paper, I suggest the results to highlight the pattern of streetscapes and the relationship between streetscapes and mental health is sufficient.

2. In the section of introduction, related literature about the relationship between streetscapes and mental health in recent year should be added, particularly those studies from Wu Jiayu. Also, studies on street view images and deep learning from Zhang Fan are also to be cited.

3. In the section of study area, the issue whether the city - Zhanjiang has peculiarities in streetscapes (street greenness and street enclosure) and human mental health should be explained.

4. Figure.4 should be improved.

5. Citations in line 225 and line 226 should be checked.

6. The section of discussion can be deepened. One possible improvement for the section is comparing these findings in this paper with existing studies. The other possible improvement is adding more detailed explanation for the results from perspective of spatial planning, urbanization, or urban development history. Also, the limitation of absence to consider social group characteristics in analyzing the association between streetscapes and mental health can be included.

7. English in this paper should be checked though the paper, for example, “published literature” in line 273, “at odds with” in line 285, “take measures” in line 304.

Author Response

Comment 1:

This paper used Baidu Street view images and deep learning to analyze the association between streetscapes and mental health in Zhanjiang, China. The topic is interesting and fits the realm of the special issues. However, this paper has many issues to be cracked as following:

Response:

We sincerely thank the reviewer for being interested in this manuscript and for providing valuable comments and suggestions. We try our best to resolve some issues and responded point-by-point.

Comment 2:

In the section of Abstract, the context of the association between streetscapes and mental health is suggested to be directly introduced. The limitation of existing studies should be explained. Also, results are overemphasized in the section, while the discussion and contribution of this paper are not introduced yet. Due to the main theme "the association between streetscapes and mental health" entitled in this paper, I suggest the results to highlight the pattern of streetscapes and the relationship between streetscapes and mental health is sufficient.

Response:

Thanks for your comments.

In the Section “Abstract”, we carefully revised this section. We only kept results about the pattern of streetscapes and the relationship between streetscapes and mental health. Please see Lines 12-26.

In section “Introduction”, We have collected and read more literatures related to streetscapes, green space and mental health. We summarized the limitation of existing studies and contributions of our study. “We find that there are still two shortcomings in the current research. First, the traditional methods using questionnaire- or GIS-based data to measure the features of the macro-scale streetscape, cannot fully examined the eye-level streetscape features. Second, most previous studies have focused on the relationship between green space and mental health, but research on the relationship between street enclosure and mental health is rare.” Please see Lines 104-110.

Comment 3:

In the section of introduction, related literature about the relationship between streetscapes and mental health in recent year should be added, particularly those studies from Wu Jiayu. Also, studies on street view images and deep learning from Zhang Fan are also to be cited.

Response:

Thanks for your comment. We have added some related works. For example, papers about the relationships between streetscapes and mental health: " Ta et al.[24]found that small green space along the road contributes to individuals' travel satisfaction and well-being. Yue et al. [25] showed that streetscape grasses have a stronger association with the mental health of older adults than street trees." in lines 60-63, literature about street view image and deep learning: " Rundle et al[34] compared seven environmental variables extracted through field surveys and Google GSV for 37 New York City neighborhoods. The results showed a high degree of consistency in 54.3% of the items. Zhang et al.[35]used street view images and deep learning to investigate the relationship between street visual elements and people’s perceptions." in lines 95-100.

We have carefully read the references you provided and added them to the manuscript. Please see Lines 36-109 and Pages 2-4.

Reference:

24 Ta, N., et al., The impact of green space exposure on satisfaction with active travel trips. Transportation Research Part D: Transport and Environment, 2021. 99: p. 103022.

25 Yue, Y., D. Yang, and D. Van Dyck, Urban greenspace and mental health in Chinese older adults: Associations across different greenspace measures and mediating effects of environmental perceptions. Health & Place, 2022. 76: p. 102856.

34 Rundle, A.G., et al., Using Google Street View to Audit Neighborhood Environments. American Journal of Preventive Medicine, 2011. 40(1): p. 94-100.

35 Zhang, F., et al., Measuring human perceptions of a large-scale urban region using machine learning. Landscape and Urban Planning, 2018. 180: p. 148-160.

Comment 4:

In the section of study area, the issue whether the city - Zhanjiang has peculiarities in streetscapes (street greenness and street enclosure) and human mental health should be explained.

Response:

Thank you for your comment. We have added some contents about urban development, urban green space and the government’s support for mental disease in Zhanjiang. We have modified the section of the study area.

"The study area is located in Zhanjiang City, between 20°12’ and 21°35’N latitude and 109°31’and 110°55’E longitude, with a total population of around 7.0 million in 2020. With its tropical monsoon climate, the average annual temperature is about 23° and the coldest monthly average temperature is 17.2°. Due to the high aesthetic quality of the urban green spaces, Zhanjiang was declared a "National Garden City" by the Chinese Ministry of Construction in 2005[37]. In 2020, the gross domestic product (GDP) was about 6.46 trillion yuan (RMB) [38]. With rapid economic development and population growth, Zhanjiang has experienced rapid urban expansion and large areas of agricultural land have been converted to nonagricultural and urban land [39]. At the same time, the Zhanjiang government has added mental diseases into their health insurance coverage to support the development of local mental health services since 2015 [40]. The study mainly covers four municipal districts of Zhanjiang, namely Chikan District, Xiashan District, Potou District, and Mazhang District” Please see Lines 122-138.

Reference:

37 Cheng, X.-L., et al., Drivers of spontaneous and cultivated species diversity in the tropical city of Zhanjiang, China. Urban Forestry & Urban Greening, 2022. 67: p. 127428.

38 Cheng, X.-L., et al., Using SPOT Data and FRAGSTAS to Analyze the Relationship between Plant Diversity and Green Space Landscape Patterns in the Tropical Coastal City of Zhanjiang, China. Remote Sensing, 2020. 12(21).

39 Cheng, X.-L., et al. Using SPOT Data and FRAGSTAS to Analyze the Relationship between Plant Diversity and Green Space Landscape Patterns in the Tropical Coastal City of Zhanjiang, China. Remote Sensing, 2020. 12, DOI: 10.3390/rs12213477.

40 Qin, X. and C.-R. Hsieh, Understanding and Addressing the Treatment Gap in Mental Healthcare: Economic Perspectives and Evidence From China. INQUIRY: The Journal of Health Care Organization, Provision, and Financing, 2020. 57: p. 0046958020950566.

Comment 5:

Figure.4 should be improved.

Response:

Thanks for your comment. The Figure 4 has been modified as below:

Comment 6:

Citations in line 225 and line 226 should be checked

Response:

Thank you for your comment. We have corrected the misquotes.

Comment 7:

The section of discussion can be deepened. One possible improvement for the section is comparing these findings in this paper with existing studies. The other possible improvement is adding more detailed explanation for the results from perspective of spatial planning, urbanization, or urban development history. Also, the limitation of absence to consider social group characteristics in analyzing the association between streetscapes and mental health can be included.

Response:

Thank you for your comments.

 In the Section “Discussion”, we added some related works about the relationships between street enclosure and mental health and compared them with our findings. For example, “Our finds are similar to those of Wang et al. [60] who measured walkability according to the visual enclosure of a neighborhood for assessing the mental health impacts. The results showed that the proportion of sky contributed to the alleviation of anxiety and depression, especially for disadvantaged older adults.” Please see Lines 406-411.

We also add some contents to understand our findings from the perspective of urban planning. For example, " As the complex impact of urbanization on health has been increasingly recognized [61], our findings may provide useful information for the municipal government to design a more livable and healthy urban space. For example, future urban environmental management projects should consider how to lower the enclosure of city streets. For example, constructing low buildings and planting grass or shrubs instead of tall trees should be considered, which is helpful to encourage people to walk more and alleviate people's stress and anxiety.” Please see Lines 413- 422.

Finally, we listed the limitations of this study, such as not considering social group characteristics.  “Furthermore, we did not take into account other social group characteristics such as education level, income, frequency and duration of street activities” Please see Lines 435-437.

60.Wang, R., et al., The relationship between visual enclosure for neighbourhood street walkability and elders’ mental health in China: Using street view images. Journal of Transport & Health, 2019. 13: p. 90-102.

61.Desai, N.G., et al., Urban mental health services in India: how complete or incomplete? Indian J Psychiatry, 2004. 46(3): p. 195-212.

Comment 8:

English in this paper should be checked though the paper, for example, “published literature” in line 273, “at odds with” in line 285, “take measures” in line 304.

Response:

Thanks for your comment. We have thoroughly checked the grammar errors and corrected them.

Reviewer 2 Report

Title: Suggest restructuring the title.

Abstract

1)    Check some spelling - hotspot @ hotspot?

2)    Missing part info on the population since mentioning some point in the conclusion. Also, it is stated in the last paragraph of the Introduction on the study’s aim.

3)    Add an association term in the abstract since it was mentioned in the title.

4)    Add GVI

5)    The study stated 2 types of methods – global and hotspot, but the abstract mentions the Spatial autocorrelation and hotspot analysis methods. Which is correct?

6)    Mention 4 districts.

 Introduction

1)    Check on the citation format and spacing before the bracket. Some citations don’t have year - Li et al./Aasgarzadeh et al/Baran et al. but later write at the end.

2)    Also, Green View index(GVI) - write properly

3)    Restructure the sentences -  However, GSV cannot be obtained in China because of strategic and business policies. Some Chinese companies, such as Baidu, have launched urban street view services (Chen et al., 2019). We thus used Baidu Street View(BSV) image in this study.

Material and Method

1)    Describe properly Figure 1. Standardize the usage Fig or Figure.

2)    State the reason why choosing 4 municipal districts?

3)    2.2 - Street View Images Data - is it supposed to be bold and full stop?

4)    Baidu Maps (https://map.baidu.com/) - write in citation style.

5)    From Fig. 2, we found that most samples are located in Chikan District and Xiashan District which is the central part of Zhanjiang. It may be because the information on the Baidu map is only available in the city center. – Since mentioning focus on 4 municipal districts, how about the readiness/balancing of the available sample to compare with the patients’ number in the hospital?

6)    Change the caption for Figure 3 to be more meaningful.

7)    Explain the buffer of (200 meters, 500 meters, 750 meters, 1000 meters) method – what is the justification for using each value.

8)    Figure 4 framework – obtaining sample has 2 flows – my concern on the flow itself seems that separate and which image need to go to which flow?  Also stated in paper “To extract SVF, panorama images were required to convert to fisheye images first”.

9)    Check on “Error! Reference source not found”

10) Is it possible to use the abbreviation for Global Moran’s Index?

11) Check the wrong Fig name in the description.

12) Distribution map of GVI hotspots – possible to write the districts to ease the capture.

13) Table 2 – I think that should put the legend for the ** as in Table 3.

14) Table 3 – change the meter to m, put the label for age and sex – arrange the title properly

Discussion

1)    What is the result comparison among the 4 districts?

Conclusion

1)    Add the future study proposal as in the discussion.

Check the references and citation format.

-        Wendel-Vos, H. S. J. P. I. M. H. K. S. d. V. T. H. W. (2016) 

Author Response

Responses to Reviewer #2

We sincerely thank the reviewer for being interested in this manuscript and for providing valuable comments and suggestions. We try our best to resolve some issues and responded point-by-point.

Title: Suggest restructuring the title.

Response:

We have collected and read more literatures related to streetscapes, green space and mental health. We find that there are still two shortcomings in the current research. First, the traditional methods using questionnaire- or GIS-based data to measure the features of the macro-scale streetscape, cannot fully examined the eye-level streetscape features. Second, most previous studies have focused on the relationship between green space and mental health, but research on the relationship between street enclosure and mental health is rare. Therefore, the aims of this study are as follows: (1) extracting two streetscape features, namely GVI and the street enclosure, using BSV images and a deep learning method, and (2) exploring the relationship between the features of urban streetscape and people's mental health by age and sex. Thus, we chose the title of the article as" Investigating the association between streetscapes and mental health in Zhanjiang, China: Using Baidu Street view images and deep learning”

Abstract:

  • Check some spelling - hotspot @ hotspot?

Response:

Thanks for your comment. We have checked all the spelling. Referring to (Wang et al., 2021) we used the term of “hotspot analysis” in this study.

Wang, X., Zhang, Y., Zhang, J., Fu, C., & Zhang, X. (2021). Progress in urban metabolism research and hotspot analysis based on CiteSpace analysis. Journal of Cleaner Production, 281, 125224. https://doi.org/https://doi.org/10.1016/j.jclepro.2020.125224

  • Missing part info on the population since mentioning some point in the conclusion. Also, it is stated in the last paragraph of the Introduction on the study’s aim?

Response:

Thanks for your comment. We have added the population info to the Abstract section.

  • Add an association term in the abstract since it was mentioned in the title.

Response:

Thanks for your comment. We have checked and modified the Abstract part.

  • Add GVI

Response:

Thanks for your comment. We have checked and modified the Abstract section.

  • The study stated 2 types of methods – global and hotspot, but the abstract mentions the Spatial autocorrelation and hotspot analysis methods. Which is correct?

Response:

Thanks for your comment. We only kept the Global Moran’s I and hotspot analysis.

  • Mention 4 districts.

Response:

Thanks for your comment. In this study, the reason why we mentioned the four districts is that we want to show the spatial distributions of GVI and streetscape in a better way. We do not intend to analyze the difference between these four districts in Zhanjiang.

Response for the whole abstract part

Thanks for your comment. We have modified the abstract to “Mental health is one of the main factors that significantly affect one’s life. Previous studies suggest that streets are the main activity space for urban residents and have important impacts on human mental health. Existing studies, however, have not fully examined the relationships between streetscape characteristics and people’s mental health on a street level. This study thus aims to explore the spatial patterns of urban streetscape features and their associations with residents’ mental health by age and sex in Zhanjiang, China. Using Baidu Street View(BSV) images and deep learning, we extracted the Green View Index (GVI) and the street enclosure to represent two physical features of the streetscapes. Global Moran’s I and hotspot analysis methods were used to examine the spatial distributions of streetscape features. We find that both GVI and street enclosure tend to cluster, but show almost opposite spatial distributions. The Results of Pearson's correlation analysis show that residents' mental health does not correlate with GVI, but it has a significant positive correlation with the street enclosure, especially for men aged 31 to 70 and women over 70-year-old. These findings emphasize the important effects of streetscapes on human health and provide useful information for urban planning.” in lines 12-23, page 1.

 Introduction

1)    Check on the citation format and spacing before the bracket. Some citations don’t have year - Li et al./Aasgarzadeh et al/Baran et al. but later write at the end.

2)    Also, Green View index(GVI) - write properly

3)    Restructure the sentences -  However, GSV cannot be obtained in China because of strategic and business policies. Some Chinese companies, such as Baidu, have launched urban street view services (Chen et al., 2019). We thus used Baidu Street View(BSV) image in this study.

 Response:

Thank you for your suggestions for the Introduction part. We have carefully checked the citations and the spellings. Referring to (Li et al., 2015) ,we used the term of “Green View index(GVI)” and abbreviation “GVI ” in this study. We have modified these sentences to “However, GSV cannot be obtained in China because of strategic and business policies. We thus used Baidu Street View(BSV), a GSV-like product developed by the Chinese company Baidu (Chen et al., 2019)”. 

Li, X., Zhang, C., Li, W., Ricard, R., Meng, Q., & Zhang, W. (2015). Assessing street-level urban greenery using Google Street View and a modified green view index. Urban Forestry & Urban Greening, 14(3), 675-685. https://doi.org/10.1016/j.ufug.2015.06.006

Material and Method

1)    Describe properly Figure 1. Standardize the usage Fig or Figure.

Response:

Thank you for your comment. We changed the name of Figure 1 to “The map of four districts of the study area”.

2)    State the reason why choosing 4 municipal districts?

Response:

Because there are four municipal districts in Zhanjiang City. The reason why we mentioned the four districts is that we want to show the spatial distributions of GVI and streetscape in a better way. We do not intend to analyze the difference between these four districts in Zhanjiang.

3)    2.2 - Street View Images Data - is it supposed to be bold and full stop?

Response:

Thanks for your comment. We changed the text to bold.

4)    Baidu Maps (https://map.baidu.com/) - write in citation style

Response:

Thank you for your suggestions. We linked the text to the Baidu website. “We obtained street view images of each sample as panorama images from the public API interface of Baidu Maps (https://map.baidu.com/), accessed on 20 December 2020.” Please see lines

5)    From Fig. 2, we found that most samples are located in Chikan District and Xiashan District which is the central part of Zhanjiang. It may be because the information on the Baidu map is only available in the city center. – Since mentioning focus on 4 municipal districts, how about the readiness/balancing of the available sample to compare with the patients’ number in the hospital?

Response:

Thank you for your comment. This is one of the limitations of this. We do not consider the balancing of the available sample of the patients’ number in four districts. We mention it the Discussion part “we do not consider the balance of the available samples of the patients in the four districts”. Please see lines 424-425.

6)    Change the caption for Figure 3 to be more meaningful.

Response:

Thanks for your comment. We have modified it to “Number of patients in different age and gender groups”.

7)    Explain the buffer of (200 meters, 500 meters, 750 meters, 1000 meters) method – what is the justification for using each value.

Response:

Thanks for the comments. We added the explanations about the buffer distance selection, such as “We built a proxy for exposure with four distances (200 m, 500 m, 750 m, and 1000 m) to streetscape according to previous studies about environmental characteristics [44, 45] and the sample distance in this study” in lines 133-135, page 5.

Reference:

  1. Ojeda Sánchez, C., et al., Urban green spaces and childhood leukemia incidence: A population-based case-control study in Madrid. Environmental Research, 2021. 202: p. 111723.
  2. Qi, M. and S. Hankey, Using Street View Imagery to Predict Street-Level Particulate Air Pollution. Environmental Science & Technology, 2021. 55(4): p. 2695-2704.

8)    Figure 4 framework – obtaining sample has 2 flows – my concern on the flow itself seems that separate and which image need to go to which flow?  Also stated in paper “To extract SVF, panorama images were required to convert to fisheye images first”.

Response:

Thank you for your comment. Figure 4 has been modified as below:

9)    Check on “Error! Reference source not found”

Response:

Thank you for your comment. All citations were checked and linked to referenced sources.

10) Is it possible to use the abbreviation for Global Moran’s Index?

Response:

Thanks for your comment. According to existing studies, it is more appropriate to use the full name of Global Moran’s Index rather than an abbreviation.

11) Check the wrong Fig name in the description.

Response:

Thanks for your comment. We have checked all the Figure names and fixed those that were wrong.

12) Distribution map of GVI hotspots – possible to write the districts to ease the capture.

Response:

Thanks for your comment. The figure was changed as follows:

 Figure 7. GVI spatial distribution map.

13) Table 2 – I think that should put the legend for the ** as in Table 3.

Response:

Thank you for the comment. We have added the Legend in Table 2.

Table 2. Pearson correlation coefficients between GVI and enclosure and the number of cases at different buffer distances.

Buffer Distance

(meter)

GVI

street enclosure

200m

-0.0582

-0.0088

500m

-0.0413

0.2484**

750m

0.014

0.3201**

1000m

0.027

0.3906**

    ** Correlation is significant at the 0.01 level (two-tailed)

14) Table 3 – change the meter to m, put the label for age and sex – arrange the title properly

Response:

Thank you for the comment. We reorganized Table 3 as following:

Table 3. Pearson correlation coefficients between GVI and enclosure and number of cases at different buffer distances and different sex and ages

Age

Sex

Street

features

Buffer distance(m)

200

500

750

1000

11-30 years old

male

GVI

-0.1812

-0.1745

-0.1396

-0.0702

street enclosure

-0.1047

0.0343

0.1042

0.1746

female

GVI

0.0886

0.0694

0.1369

0.2141

street enclosure

-0.3539

-0.01

0.2238

0.2821

31-50 years old

male

GVI

0.0762

0.0896

0.0846

-0.0153

street enclosure

0.0404

0.0494

0.2815*

0.3936**

female

GVI

-0.1911

0.1367

-0.0459

-0.0047

street enclosure

-0.036

0.0132

-0.2354

0.1907

51-70 years old

male

GVI

-0.0147

-0.2487

-0.1823

0.0873

street enclosure

-0.1004

0.322

0.4852*

0.4286*

female

GVI

-0.0167

0.0091

0.0923

0.0582

street enclosure

0.1042

0.1229

0.0461

0.2714

71-90 years old

male

GVI

NaN

0.6071

0.0614

0.1275

street enclosure

NaN

0.273

0.3869

-0.1129

female

GVI

-0.3087

-0.1526

-0.5332

0.0205

street enclosure

0.0775

0.5477

0.5716

0.7057*

** Correlation is significant at the 0.01 level (two-tailed); * Correlation is significant at the 0.05 level (two-tailed).

Discussion

  • What is the result comparison among the 4 districts?

 Response:

Thanks for your comment. We did not intend to compare the differences between the four districts in this study. But we compared them with our findings with related work about the relationships between street enclosure and mental health. We also discuss our results from the perspective of urban planning.

Conclusion

1)    Add the future study proposal as in the discussion.

Response:

Thanks for your comment. We added future research plans at the end of the discussion section “To summarize, future study should highlight these points (1) more streetscape characteristics and social group features should be considered; and (2) try to find other indicators to represent human mental health” in lines 435-439.

Check the references and citation format.

Response :

Thank you for your comment. We have checked all the citation formats.

-        Wendel-Vos, H. S. J. P. I. M. H. K. S. d. V. T. H. W. (2016) 

Round 2

Reviewer 1 Report

The authors have addressed all issues that I have mentioned. Therefore, this paper is suggesteed to be accepted in current form.